# ALGO: Synthesizing Algorithmic Programs with LLM-Generated Oracle Verifiers

**Kexun Zhang**[1]  **Danqing Wang**[1]  **Jingtao Xia**[1]  **William Yang Wang**[1]  **Lei Li**[2]
[1]University of California Santa Barbara   [2]Carnegie Mellon University
{kexun,danqingwang,jingtaoxia,william}@ucsb.edu   leili@cs.cmu.edu

## Abstract

Large language models (LLMs) excel at implementing code from functionality descriptions but struggle with algorithmic problems that require not only implementation but also identification of the suitable algorithm. Moreover, LLM-generated programs lack guaranteed correctness and require human verification. To address these challenges, we propose ALGO, a framework that synthesizes **A**lgorithmic programs with **LLM-G**enerated **O**racles to guide the generation and verify their correctness. ALGO first generates a reference oracle by prompting an LLM to exhaustively enumerate all the combinations of relevant variables. This oracle is then utilized to guide an arbitrary search strategy in exploring the algorithm space and to verify the synthesized algorithms. Our study shows that the LLM-generated oracles are correct for 88% of the cases. With the oracles as verifiers, ALGO can be integrated with any existing code generation model in a model-agnostic manner to enhance its performance. Experiments show that when equipped with ALGO, we achieve an 8× better one-submission pass rate over the Codex model and a 2.6× better one-submission pass rate over CodeT, the current state-of-the-art model on CodeContests. We can also get 1.3× better pass rate over the ChatGPT Code Interpreter on unseen problems. The problem set we used for testing, the prompts we used, the verifier and solution programs, and the test cases generated by ALGO are available at `https://github.com/zkx06111/ALGO`.

## 1 Introduction

Large Language Models (LLMs) have demonstrated significant prowess in generating code from natural language descriptions. Models such as Codex [5] and CodeGen [23] can easily achieve over 30% pass@1 accuracy on HumanEval, a docstring-to-code dataset. However, these models struggle when faced with algorithmic problems akin to those encountered in CodeContests [17]. Even with reasonable sample limits, achieving a 10% accuracy rate remains a considerable challenge [4]. More recently, the GPT-4 model, despite being provided with problem descriptions and solution hints, managed a pass@1 accuracy of only 10.6% [3] on LeetCode Hard problems.

The verifiability of LLM-based code generation presents another significant challenge. Without verifiability, code generation systems can hardly earn trust for production use. Users of GitHub Copilot [21], an AI assistant for programmers, reportedly spend over 20% of their time verifying the suggested code snippets. Existing approaches toward verification and reranking LLM-generated programs either rely on neural models to predict confidence scores [22, 32] or require LLM-generated test cases to execute the programs [4, 15]. However, neural verifiers do not provide interpretable feedbacks and LLM-generated test cases are often incorrect.

Traditional code synthesis techniques rely on *oracles* for the verification of the synthesized code. These oracles are typically derived from formal specifications [20, 25, 28, 29]. They can also be employed in program synthesis under the widely-used framework of oracle-guided inductive synthesis (OGIS) [14]. Despite their widespread application in code verification and synthesis, the

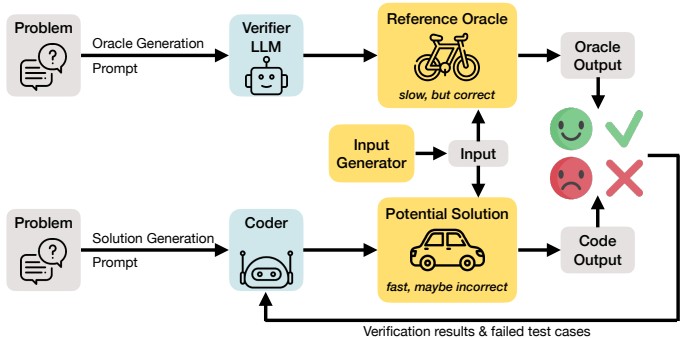

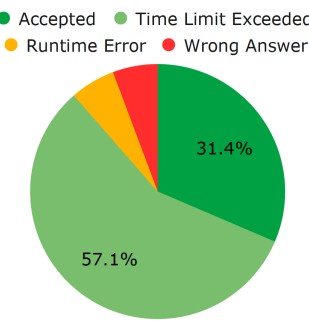

Figure 1: The ALGO pipeline. The verifier LLM generates the probably correct but possibly slow reference oracle that solves the problem with an exhaustive search. The coder generates a more efficient candidate program and refines the candidate program by comparing its output with the oracle's output. The coder can be any existing code generation model.

Figure 2: The reference oracle generated by ALGO is correct for 88.5% of the problems in our benchmark. We consider an oracle correct if it gets *Accepted / Time Limit Exceeded* verdict on LeetCode website and is checked by human experts as a reasonable solution.

use of oracles often demands heavy human involvement, and the automatic generation of oracles is challenging. Existing strategies focus on generating oracles that can only detect crashes (safety oracle) or bugs introduced by future changes (regression oracle), rather than semantic bugs crucial for algorithmic programs [9, 18, 24].

Motivated by the potential benefits of oracles in traditional code synthesis, we introduce ALGO, a framework that leverages oracles generated by large language models to address the challenges in LLM-based code synthesis and verification. As illustrated in Figure 1, ALGO incorporates two modules for algorithm synthesis. The verifier is instructed to generate an exhaustive search algorithm regardless of time efficiency, thus acting as a reference oracle. Concurrently, the coder is asked to find a more efficient solution with any prompts or search strategies. The candidate program's correctness is then evaluated by comparing its output with that of the oracle for a given set of test inputs. The results of the verification, along with any test cases where the candidate program falls short, are subsequently provided to the coder for code refinement.

We evaluated ALGO based on two key dimensions: the oracle's verification capability and its potential to enhance algorithm synthesis. Our experiments reveal that the reference oracles are correct for 88.5% of the problems (as shown in Figure 2). Moreover, the oracles' verdicts are in agreement with the golden verdicts of the online judge 75% of the time. To examine the enhancement in accuracy offered by ALGO, we integrated it with several existing code generation models including Codex [5], CodeT [4], PG-TD [31] and ChatGPT Code Interpreter [1]. We observed that ALGO significantly boosts their performance on CodeContests [17] and a collection of recent LeetCode problems. Our experiments show that ALGO notably enhances the performance of these models: in terms of one-submission pass rate, Codex's performance improved by a factor of 8, CodeT's by 2.6, PG-TD's by 1.5, and ChatGPT Code Interpreter's by 1.3.

Our contributions can be summarized as follows:

- We present ALGO, a novel framework for **A**lgorithm synthesis that utilizes **LLM**-**G**enerated reference **O**racles as verifiers. It is a model-agnostic framework that can be integrated with any code generation model to verify candidate solutions with reliable test cases generated by reference oracles.
- We conduct a comprehensive evaluation of synthesis accuracy and verifiability of ALGO, utilizing several distinct code generation models in a versatile, model-agnostic manner. Our results indicate that ALGO can generate high-quality oracles and test cases that lead to significant improvements in code generation.

---

[1]ChatGPT Code Interpreter is a variant of GPT-3.5 with the code interpreter plugin. The version we evaluated, released on March 24th, utilizes the underlying model `text-davinci-002-code`. It is now offline and replaced with `gpt-4-code-interpreter`.

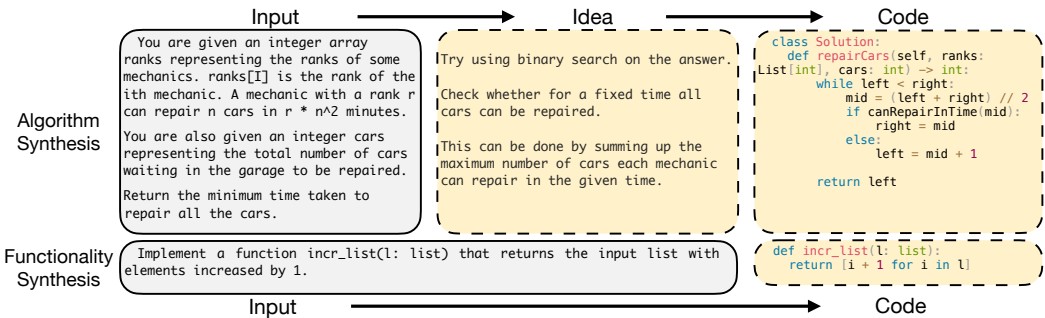

Figure 3: Examples of algorithm synthesis (top) and functionality synthesis (bottom). The parts in solid boxes are given by the problem, while the parts in dotted boxes should be inferred by the model. Functionality synthesis does not require the model to infer the idea, while algorithm synthesis does.

## 2    Algorithm Synthesis

Large language models excel at generating simple programs. For example, ChatGPT can easily achieve over 70% pass@1 accuracy on the docstring-to-code dataset HumanEval. However, it still struggles with complex algorithmic problems such as CodeContests. We hypothesize that this is because algorithmic problems need efficient solutions rather than straightforward and brute-force ones. To test our hypothesis, we prompt ChatGPT in two different ways to see if they result in different success rates. One prompt directly describes the problem and asks the model to generate a solution that meets the time limits. The other asks the model to ignore the time limit and to solve the problem in the most straightforward way. The prompts we used are listed in Appendix A.2. We evaluate the correctness of the brute-forces and efficient solutions both manually and using official test cases without time limits so that a correct yet inefficient program is also considered a success.

Table 1: The success rates when ChatGPT is prompted to generate efficient solutions and brute-force solutions and their relative differences. Generating brute-force solutions for LeetCode and CodeContests is much easier than generating efficient solutions.

|                                | LeetCode | CodeContests | HumanEval |
|--------------------------------|----------|--------------|-----------|
| Brute-force success rate       | 88.5%    | 72.0%        | 70.1%     |
| Efficient solution success rate| 41.2%    | 7.90%        | 72.5%     |
| Relative difference            | 114%     | 822%         | -3%       |

The results in Table 1 validate our hypothesis. For LeetCode and CodeContests, it is much easier to generate brute-forces than efficient solutions. But for HumanEval, the two success rates are similar. This clearly indicates that problems in HumanEval and those in LeetCode and CodeContests are of two different types. The latter is more difficult to solve because it requires the application of algorithms. Based on that, we divide code synthesis into two categories: *functionality synthesis* and *algorithm synthesis*. Typically, *functionality synthesis* problems, such as those seen in HumanEval [5] and MBPP [1], provide a detailed description that can be readily translated into an implementation. On the other hand, *algorithm synthesis* problems found in APPS [11] and CodeContests [17] are more abstract and need efficient algorithms to solve. In this paper, we focus on addressing *algorithm synthesis* rather than *functionality synthesis*. An algorithmic synthesizer, thus, has an additional task to that of a functionality synthesizer—it needs to generate the solution idea, implicitly or explicitly, before synthesizing the code. We further clarify the contrast between the two by providing typical examples of each in Figure 3.

**Formal definition of algorithm synthesis and functionality synthesis.** A code generation task can be defined as a tuple $(Q, J) \in \mathcal{Q} \times \mathcal{J}$, where $Q$ is the problem description and $J : \mathcal{P} \to \texttt{Bool}$ is the system judge. The system judge $J$ evaluates if a program from the program space $\mathcal{P}$ solves the problem represented by $Q$. The goal of the synthesizer is to produce a program $P \in \mathcal{P}$ that $J(P) = \texttt{True}$. The system judge $J$ is further composed of two components: $J_S : \mathcal{P} \to \texttt{Bool}$ and

Figure 4: The prompt we used for oracle generation and one oracle generated with it. The instructions are in blue. The correct solution solves the problem with binary search in polynomial time, while the generated oracle takes exponential time to enumerate all possible work allocations.

$J_T : \mathcal{P} \to$ `Bool`. $J_S$ checks the semantic correctness of the generated code, while $J_T$ ensures that the efficiency of the generated code satisfies the requisite conditions. Intuitively, $J(P) = J_T(P) \land J_S(P)$.

## 3 Algorithmic Synthesis with LLM-Generated Oracles (ALGO)

As discussed in Section 2, generating brute-force solutions to algorithm synthesis tasks is much easier than generating efficient solutions; this huge gap in difficulty can be exploited. To solve algorithm synthesis, we propose Algorithmic Synthesis with LLM-Generated Oracles (ALGO). As shown in Figure 1, the ALGO framework utilizes two components - a **coder** and a **verifier** - to solve algorithmic problems. The **coder** takes the problem description $Q$ and optionally, the verification results from its last generation and generates a program $P$ that solves $Q$. The **verifier** generates a reference oracle whose outputs are used to verify whether the candidate program generated by the coder is correct. For each problem, ALGO creates the verifier once and uses it to guide arbitrary coders, allowing ALGO to be model-agnostic.

### 3.1 Verification with Oracle

**Oracle Generation** The difference between a reference oracle $P_O$ and an actual solution $P_G$ is that $P_O$ only needs to be semantically correct (i.e. $J_S(P_O) =$ `True`) while $P_G$ needs to be correct and efficient (i.e. $J(P_G) = J_S(P_G) \land J_T(P_G) =$ `True`). We utilize this difference to generate the oracle. When there is no time limit, most algorithmic problems can be solved with an exhaustive search [8]. As depicted in Figure 4, we put the process of an exhaustive search algorithm (which is the same for every problem) in the prompt along with the problem description. LLMs, which excel at implementing a program when provided with clear instructions, are then able to generate reference oracles.

**The Verification Process** To handle a verification request, we must return a verdict of `True/False` regarding $P$'s correctness, and optionally, the inputs that cause $P$ to fail. We utilize an **input generator** program to create random test inputs in line with problem constraints. These test inputs are then supplied to both the program under verification $P$, and the oracle $P_O$. The test outputs of $P$ are compared against those of $P_O$. If they match, a `True` verdict is returned. If they don't, the failed test cases are returned to the synthesizer along with a `False` verdict.

### 3.2 Code Synthesis Strategies

ALGO's coder can use different strategies for code synthesis. The code synthesis process can either be one-time or iterative, depending on the used coder. During a one-time synthesis, the coder generates a batch of solutions to the problem, and these solutions are verified and ranked according to their verification results. During an iterative synthesis, the code generates solutions, gets their verification results (and optionally mistaken test cases), and then revises the solutions iteratively.

Other than the coder model itself, its strategy to search for and identify the suitable algorithm for the problem also affects the synthesis accuracy significantly. The coder's search strategy can be defined as a conditional distribution $\pi(P|Q)$ of the program $P$ conditioned on the problem description $Q$. To generate code is to sample programs from this distribution. To compute this distribution, we can define a space of latent variables $\mathcal{I}$ and marginalize over it, as denoted by

$$\pi(P|Q) = \sum_{I \in \mathcal{I}} \pi(P|I)\pi(I|Q).$$

ALGO is a versatile framework that can incorporate a variety of search strategies. A strategy may be implicit, with the variable $I$ not specified, or explicit, specifying $\mathcal{I}$, the space of $I$, and the methodology to traverse this space. Here, we give three examples: *an implicit searcher*, *an instruction enumerator*, and *an iterative searcher*. However, it is crucial to note that these examples are representative, and there are numerous alternative search strategies in ALGO.

**Implicit Searcher:** An implicit searcher samples a program directly from the distribution determined by the underlying model, using the problem description. It solely relies on the ability of the code generation model and depends on the oracle to filter the generated programs post hoc.

**Instruction Enumerator:** An instruction enumerator is an explicit searcher where $\mathcal{I}$ is a set of pre-defined instructions. Since the performance of instruction-tuned LLMs varies significantly based on the instructions they are given, it is natural to consider enumerating a space of instructions to get the best result. An instruction enumerator first picks an instruction from a pre-defined instruction set $\mathcal{I}$, and then synthesizes the program by instructing the language model with $I$. In our example of an instruction enumerator, we choose $\mathcal{I}$ to be a set of high-level ideas of algorithms such as '*Binary Search*' and '*Sorting*'. We first sample possible solution algorithms and then instruct the large language models to generate programs with the ideas given.

**Iterative Searcher:** A iterative searcher is an explicit searcher that takes the signal from the verifier to refine its output. The searcher space $\mathcal{I}$ can be the token vocabulary of the model. It uses a search algorithm to determine the next token during the decoding process by enumerating the vocabulary tokens given the problem description and the partial program in each generation step. Since the search space is exponentially large, it would be time-intensive. Usually, the searcher is guided by some rewards from the verifier to prune the candidate space in each step.

## 4 Experiments

In this section, we implement ALGO with three code synthesis strategies, and evaluate it with two challenging algorithmic benchmarks to validate its flexibility and effectiveness. Moreover, we investigate the verifier's performance by evaluating the quality of the generated reference oracle and test cases.

### 4.1 Experiment Setup

**Verifier** We employ ChatGPT Code Interpreter to create the verifier. It is first prompted to generate the reference oracle and then the input generator. As mentioned in Section 3.1, this is possible because the reference solutions to most algorithmic problems involve exhaustively going through all possible solutions to find a feasible or optimal one. We use a temperature of 1.0 and resample the solution until it can pass all example cases. The prompt we used for generating the oracle is shown in Figure 4. For the verifier, we simply use zero-shot prompts to generate the input generator with an input validator since both of them are functionality syntheses and easy to generate. We set the maximum length of each input variable to 10 and skip out cases that raise the timeout exception and the recursion error when generating the output via the reference oracle. The prompts we used are listed in Appendix A.2 with generation examples. Note that for each problem, we only need to create the verifier once, and it can be used to create arbitrary test cases and guide arbitrary models.

**Code Synthesis Strategies** We integrate the following baselines as the coder with ALGO to evaluate its verification capability and synthesis accuracy:

• **Codex** [5] and **CodeT** [4]: Codex and CodeT are used as coders that utilize *implicit searchers*. Codex is a specialized language model, trained on publicly available code from GitHub. CodeT uses Codex to generate both programs and test cases for a problem and uses their dual agreement to

filter and rank the programs generated. To evaluate ALGO, we use the exact same set of programs Codex generated, and rank them in three different ways: in a random manner, based on the dual agreement heuristics from CodeT, and according to the verification results provided by ALGO. We then compare the top-ranked programs from each of these three rankings. Note that due to OpenAI's policy, the Codex model is no longer available so we directly use the Codex-generated programs provided by the CodeT paper[2].

- **ChatGPT Code Interpreter**: ChatGPT Code Interpreter is used as a coder that utilizes *instruction enumerators*. It is a variant of GPT-3.5 with the code interpreter plugin. The version we evaluated, released on March 24th, utilizes the underlying model `text-davinci-002-code`. As the Code Interpreter is instruction-tuned, it can follow directives to solve a problem using a specific class of algorithms, such as 'Binary Search'. This capability allows it to employ a search strategy that iterates through a list of algorithm categories to identify a correct solution. Moreover, with the code interpreter plugin, it is able to run Python code and interpret its results, and automatically refine its output based on the program output. We use the default temperature of 1.0 and resample the code solution until it can pass the example cases in the problem descriptions or reach the resample limit (N=5).
- **PG-TD** [31]: PG-TD is used as a coder that utilizes *iterative searchers*. In contrast to CodeT, which employs test cases post hoc, PG-TD incorporates them during the decoding process. It implements a tree search-based planning algorithm for decoding, utilizing the reward from test cases to estimate the value of the children nodes in the search tree. PG-TD uses GPT-2 [26] and GPT-NEO [2] as the underlying model. We implement two versions: PG-TD* is only based on the public test cases to generate programs, and PG-TD has access to both public test cases and the generated test cases by Li et al. [17]. Both are based on the GPT-2 fine-tuned on the APPS training dataset, which was released by Zhang et al. [31]. We integrate PG-TD into ALGO by substituting the generated test cases with the reward from the verifier.

**Benchmarks**    We utilize the following benchmarks to evaluate ALGO:

- **CodeContests** [17]: We evaluate ALGO with Codex, CodeT and PG-TM on all 165 problems from CodeContests test set, which are collected from Codeforces, a competitive programming website. All the three baselines we evaluate have been evaluated on CodeContests in their own paper, we follow the exact same setting to ensure a fair comparison.
- **LeetCode**: We evaluate ALGO with ChatGPT Code Interpreter on 35 recently released LeetCode problems. To avoid the influence of contamination and faithfully evaluate ALGO's ability in verification and guiding code synthesis, we made sure these problems are released concurrently or after the release of GPT-4. We manually annotate the algorithm categories for the solutions to these problems to evaluate the search strategy of enumerating algorithm categories. The list of problems and their corresponding categories is listed in Appendix A.3.

**Metric**    Following previous studies [31, 4], we use the $n@k$ metric [16, 5, 17] to evaluate the accuracy of code synthesis. To compute $n@k$ for a problem, a code synthesizer needs to generate $k$ candidate programs and select the top-$n$ candidates to submit according to some reranking mechanism, which in our case is the LLM-generated oracles and test samples. $n@k$ is the proportion of problems that can be solved by any of the top-$n$ submissions. Since some code synthesizers do not have reranking mechanisms, we define their $n@k$ as the proportion of problems solved with $n$ submissions randomly sampled from $k$ candidates. To ensure a fair comparison, we use the same sampling budget $n$ under the same setting when evaluating the performance of ALGO and baselines.

### 4.2    Synthesis Accuracy

**CodeContests**    We present ALGO's $n@k$ accuracy for CodeContests in Table 2. In the first part of the table, Codex is used as an implicit searcher. Compared to Codex itself and CodeT which also heuristically evaluates the program Codex generates, ALGO consistently outperforms them at every value of $n$. The most significant performance enhancement by our method is observed at $1@k$, yielding an 8× improvement over Codex, and a 2.6× improvement over CodeT. As the value of $k$ increases, the performance advantage diminishes. This trend is expected, as the chosen sample number $n$ approaches the total sample count $k$, causing the pass rate to converge towards the ratio of correct programs within the total program set. The same phenomenon is also observed in the second

---

[2]https://github.com/microsoft/CodeT/

Table 2: Performance on CodeContests. Gain is the performance improvement compared with the baseline (the first line in each part). ALGO significantly outperforms other baselines, especially for $1@k$. Note that $k = 20$ for ChatGPT-based settings, while $k = 1000$ for other settings.

| Method | $1@k$ | Gain | $2@k$ | Gain | $10@k$ | Gain | $100@k$ | Gain |
|---|---|---|---|---|---|---|---|---|
| Codex-Based One-Time Synthesis ($k = 1000$) | | | | | | | | |
| Codex [5] | 0.70 | - | 1.20 | - | 3.00 | - | 7.50 | - |
| CodeT [4] | 2.10 | +1.40 | 2.30 | +1.10 | 5.30 | +2.30 | 9.90 | +2.40 |
| ALGO w/ Codex | **5.60** | **+4.90** | **5.60** | **+4.40** | **7.70** | **+4.70** | **11.10** | **+3.60** |
| GPT-2-Based Iterative Synthesis ($k = 20$) | | | | | | | | |
| PG-TD* | 0.62 | - | 1.07 | - | 2.26 | - | 3.45 | - |
| PG-TD [31] | 0.67 | +0.05 | 1.12 | +0.05 | 2.53 | +0.27 | 3.81 | +0.36 |
| ALGO w/ PG-TD | **1.57** | **+0.95** | **2.44** | **+1.37** | **3.67** | **+1.41** | **4.06** | **+0.61** |
| ChatGPT-Based One-Time Synthesis ($k = 20$) | | | | | | | | |
| ChatGPT | 4.40 | - | 6.62 | - | 12.21 | - | - | - |
| ALGO w/ ChatGPT | **12.00** | **+7.60** | **12.00** | **5.38** | **14.00** | **+1.79** | - | - |

and third part of Table 2, where PG-TD and ChatGPT are used as coders. ALGO is able to achieve a 2.5× improvement over PG-TD and a 2.8× improvement over ChatGPT at $1@k$.

**LeetCode** We list ALGO's one-submission pass rate on LeetCode problems in Figure 5. The instruction set we enumerate for each problem is a set with several algorithm categories, among which there is one correct category. ALGO's superior performance compared to ChatGPT and GPT-4 demonstrates the benefits of an instruction searcher that enumerates possible algorithms. It also demonstrates ALGO's ability to filter and select the correct algorithm category. Without the verifier, we would not have been able to tell right from wrong.

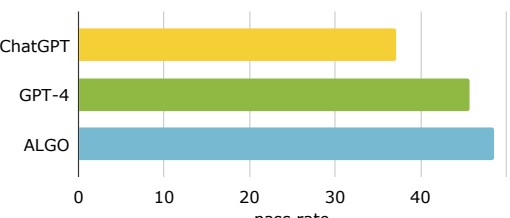

Figure 5: Performance on LeetCode. Code interpreter integrated with ALGO can surpass GPT-4's pass rate.

### 4.3 Verification Analysis

The verifier is critical to ALGO as it guides the creation of an efficient and accurate solution. In this section, we scrutinize the quality of the verifier from several perspectives.

**LLM-generated oracles are usually correct.** The correctness of LLM-generated oracles is crucial to the correctness of ALGO's test cases. We examine their correctness with both the system judge and human experts. We submitted LLM-generated oracles to the online judges (Codeforces and LeetCode) where the problems came from, to get the system verdicts. Oracles with verdicts other than *Accepted* (AC), *Time Limit Exceeded* (TLE), and *Runtime Error* (RE) were directly considered incorrect. For those with TLE and RE verdicts that could potentially be semantically correct despite exceeding the time limit or stack size, we hired an experienced competitive programming contestant to examine if they were semantically correct and not efficient enough. For LeetCode problems, as displayed in Figure 2, 88.5% of the oracles are semantically correct, including ***all*** those with TLE verdicts. For codecontests problems, 72% of the oracles are semantically correct.

**Oracle-generated test cases have better quality.** We evaluate the quality of ALGO's test cases generated by answering the following questions: (i) *Do verification verdicts from generated cases agree more with the system judge than verdicts from only the public tests?* (ii) *Do the generated tests achieve higher statement coverage compared with public tests?* In this context, we calculate two metrics: **Agreement** and **Coverage**. **Agreement** is the consistency between the decision of the system judge and the test case set. For instance, candidates that fail the system judge should also fail on the generated test set, while those that can pass the judge should also succeed on all test cases. **Coverage** is the percentage of statements in a code solution that are executed by test cases. Following

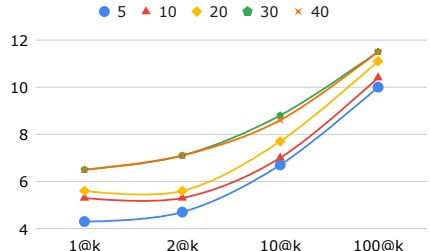

Figure 6: The performance of different sizes of test cases on CodeContests.

Table 3: The quality of test case. Compared to the example cases given in the problem description, the test cases generated by ALGO's verifier detect more failures in the programs, agree better with the system judge $J$, and covers more lines in the generated code.

| Dataset | Case | Agreement | Coverage |
|---|---|---|---|
| CodeContests | Example | 57.39% | 85.78% |
| | ALGO | 94.78% | 89.39% |
| LeetCode | Example | 68.57% | 90.32% |
| | ALGO | 74.29% | 93.00% |

Chen et al. [4], we use a public coverage library[3] to gather the coverage results. As demonstrated in Table 3, our generated test cases substantially improve the agreement with the standard judge (+37.39% on CodeContests and +5.72% on LeetCode), indicating they can enhance the verification of candidates prior to submission. In addition, test cases of ALGO achieve higher coverage of the statements, implying that these cases can more effectively validate the functionality of the solution.

**Better test cases lead to better results.** The test cases generated by ALGO have better agreement and coverage. We check if they lead to better synthesis by comparing them to the other two sets of test cases - examples provided in the problem statement and test cases directly generated by ChatGPT. We used the same coder with the exact same sample budget $k = 20$. As reported in Table 4, under the same setting, ALGO tests led to much better performance.

Table 4: Given the same coder, ALGO test cases lead to much better $n@k$ performance, indicating that better test cases do help in code synthesis.

| | 1@20 | 3@20 | 7@20 |
|---|---|---|---|
| Example Tests | 4.4% | 7.9% | 12.2% |
| ChatGPT Tests | 6.8% | 8.2% | 11.7% |
| ALGO Tests | **12.0%** | **12.0%** | **14.0%** |

**More test cases lead to better results.** We examine the impact of the size of test cases the verifier generates on CodeContests for ALGO w/ Codex. Our verifier has the flexibility to create an arbitrary number of cases and use test case sets of varying sizes to rank and guide the synthesis. As shown in Figure 6, when the size is fewer than 30, the increase in the size of test cases can significantly bolster performance for all $n@k$, indicating the effectiveness of our verifier in covering more corner cases with additional test cases. Notably, having a sufficient number of test cases is more critical for $1@k$, which relies on test cases to select the optimal solution for one-time submission. However, this improvement diminishes when $n$ is larger. This is because a robust verifier is not strongly required when the chosen sample number $n$ is large, as discussed in Section 4.2. Furthermore, there is very little additional performance gain with more than 30 test cases. We hypothesize that this is due to the quality of our oracle and the generation range we have set for the verifier.

## 4.4 Case Study

We demonstrate the workflow of ALGO with a problem from our LeetCode benchmark [4]. This problem asks for the optimal allocation of cars to mechanics with varying efficiency to minimize the total repair time, when the mechanics would work simultaneously. As depicted in Figure 7, the code generated using a reference oracle prompt exhaustively enumerates all potential assignments of cars and selects the optimal one according to the problem description. Assuming $n = \text{len}(ranks), m = cars$, this program exhibits exponential time complexity $T_O(n, m) = \mathcal{O}(2^m)$, making it an inefficient solution that is unable to pass the online judge. Nevertheless, despite this program's inefficient nature, this implementation remains valuable in terms of semantic correctness (i.e., $J_T(P_O) = \text{False}$ and $J_S(P_O) = \text{True}$, where we can conveniently define $J_T(P) = $ "time complexity of $P$ is at most $\mathcal{O}((n + m) \log (n + m))$ " without loss). This example demonstrates how ALGO can generate reference oracles for code synthesis.

---

[3] https://coverage.readthedocs.io/en/6.4.2/
[4] https://LeetCode.com/problems/minimum-time-to-repair-cars/

**Test Inputs Generated by ALGO**

Test Input $I_0$: ranks=[4,2,3,1],cars=10

$I_1$  $I_2$  $I_3$  ...

**Reference Oracle Generated by LLM**
```python
class BruteforceSolution:
  def repairCars(self, ranks: List[int], cars: int) ->
int:
    num_mechanics = len(ranks)
    min_time = float('inf')
    def generate_allocations(allocation, remaining_cars):
      nonlocal min_time
      if len(allocation) == num_mechanics:
        if sum(allocation) == cars:
          times = [ranks[i] * allocation[i] * \
            allocation[I] for i in range(num_mechanics)]
          max_time = max(times)
          min_time = min(min_time, max_time)
        return
      for num_cars in range(remaining_cars + 1):
        new_allocation = allocation + [num_cars]
        new_remaining_cars = remaining_cars - num_cars
        generate_allocations(new_allocation,\
                             new_remaining_cars)
    generate_allocations([], cars)
    return min_time
```

**Potential Greedy Solution**  **System Judge: Wrong Answer**
```python
class Solution:
  def repairCars(self, ranks: List[int], cars: int):
    ranks.sort()
    max_repair_time = 0
    assigned_cars = 0
    for rank in ranks:
      num_cars = (cars + len(ranks) - assigned_cars -
1) // (len(ranks) - assigned_cars)
      num_cars = min(num_cars, cars - assigned_cars)
      repair_time = rank * num_cars * num_cars
      max_repair_time = max(max_repair_time,
repair_time)
      assigned_cars += num_cars
    return max_repair_time
      num_cars = min(num_cars, cars - assigned_cars)
      repair_time = rank * num_cars * num_cars
      max_repair_time = max(max_repair_time,
repair_time)
      assigned_cars += num_cars
    return max_repair_time
```

100

16

**Potential Binary Search Solution**
```python
class Solution:
  def repairCars(self, ranks: List[int], cars: int) ->
int:
    def countCarsRepaired(ranks, minutes):
      total_cars = 0
      for rank in ranks:
        n = int((minutes / rank) ** 0.5)
        total_cars += n
      return total_cars
    left = 1
    right = max(ranks) * cars * cars
    while left < right:
      mid = (left + right) // 2
      if countCarsRepaired(ranks, mid) < cars:
        left = mid + 1
      else:
        right = mid
    return left
```

16

**System Judge: Accepted**

Figure 7: ALGO is capable of verifying two candidate programs, in different algorithm categories: greedy and binary search. Using an input-output example provided by the reference oracle, ALGO is able to deliver the verification result consistent with the system judgment.

In Figure 7, we present how the reference oracle can verify the correctness of candidate programs generated by the coder LLM. We present two candidate programs generated with different algorithm categories: greedy and binary search. Specifically, the binary search candidate $P_B$ is generated with instructions on binary search and the greedy candidate $P_G$ is generated without explicit guidance. The binary search algorithm is ideally suited for this problem, given that the indicator function, denoted as $f(t)$, which determines whether all cars can be repaired within $t$ minutes, is monotonous. This makes it possible to apply binary search on the time $t$ and turn the optimization problem into a judgement problem. The binary search candidate represents an implementation of such algorithmic approach, which is an efficient and correct solution of time complexity $T_B(n, m) = \mathcal{O}(n \log m)$ and can pass the system judge (i.e., $J(P_B) = \texttt{True}$). On the other hand, greedy algorithm is not suitable for this problem. Thus, even though the greedy candidate presented is of time complexity $T_G(n, m) = \mathcal{O}(n)$, the semantic of this candidate is wrong (i.e., $J_T(P_G) = \texttt{True}$ and $J_S(P_G) = \texttt{False}$). This example demonstrates that providing the correct instruction can lead to better code synthesis result.

In Figure 7, $P_G$ failed with our verifier since $P_G(I_0) \neq P_O(I_0)$. Similarly, $P_B(I_0) = P_O(I_0)$ indicates $P_B(I_0)$ passed our verifier. This verification result is consistent with the system judge and demonstrates that ALGO-generated oracles can help to verify the programs generated. Furthermore, the verification results can help in finding the correct instruction (algorithm category).

## 5 Related Work

**Reranking Techniques for LLM Code Generation.**  LLMs often need to sample many candidates before finding a single correct solution. Therefore, reranking techniques for telling the correct programs among many candidates are crucial for better LLM coders. One line of work in reranking involves neural models that predict some form of *confidence score* for program candidates. This score can be the likelihood of the program [5], the mutual information between the code and the problem description [32], or outputs from verifier models that are purposely trained to predict the correctness of programs [22, 13]. However, their scores are just scalars and not interpretable. Both the oracles and test results from ALGO are much easier to check and interpret.

Another line of work utilizes test cases and the programs' execution results to rerank them, which is much more similar to ALGO's approach. Shi et al. [27] and Li et al. [17] cluster different candidates

according to their execution results on example cases, creating a mechanism similar to majority voting. Chen et al. [4] and Key et al. [15] also cross-check programs and execution results, but they create more test cases with the coder models to make the tests stronger. However, LLM-generated test cases for a single problem can contain both correct and incorrect ones, providing harmful feedback to the coder. ALGO has converted the task of checking AI-generated programs to checking AI-generated brute-forces, which is much easier.

**Oracles in Program Synthesis and Software Testing.** In the traditional program synthesis domain, automatic verification of a synthesized program is possible [29, 28] because formal specifications, including input-output examples [25] or logical constraints [20], are explicitly provided. In contrast, oracle generation using LLMs is challenging due to the ambiguous nature of text and lack of formal specifications, making it difficult to do verification. Efforts have been made to create formal specifications from text [6, 15, 10], but reliability remains an issue. Meanwhile, in the software testing domain, test oracle generation [9, 18, 24] is a well-known and challenging task. However, existing work mainly focuses on generating regression tests, which are not intended for new bug detection. Moreover, many methods rely heavily on formatted documents or are designed for program crashes or basic functional properties, making them unsuitable for complex algorithmic problems.

**Self-Refining Language Models.** Large language models (LLMs) have exhibited a remarkable capability for self-analysis and self-improvement, as highlighted by numerous studies [19, 12, 30]. Such reflection is also used to improve the quality of the generated programs. For instance, Chen et al. [7] trains Large Language Models (LLMs) to generate explanations for code and utilizes both the explanation and the execution results as feedback to improve coding solutions. Self-Refine [19] prompts the LLM to provide feedback on the efficiency of its own code solutions and refine these solutions based on the feedback given. In this paper, we use the execution results of the reference oracle as feedback and employ them to guide the generation of programs.

## 6   Conclusion and Discussion

We proposed ALGO, an algorithm synthesis framework that leverages LLM-generated oracles as verifiers to synthesize algorithmic programs. ALGO consists of a coder and a verifier. The coder generates code solutions to the problem and requests verification from the verifier to iteratively refine its idea and its code. The verifier employs LLM-generated oracles to generate test cases for verification. ALGO can employ different types of search strategies with different underlying code generation models. We introduced a method to synthesize the reference oracle with LLMs based on the ease to generate brute-force solutions for algorithmic problems.

We extensively evaluated ALGO's performance on two different datasets with three baselines. ALGO outperformed various baselines with different search strategies by a large margin. We also did a detailed analysis of the reference oracles generated on their semantic correctness, and their verdict agreement with the system judge. The oracles in ALGO have a high accuracy and make highly consistent verification results with the system judge. The high quality of the oracles supports the performance of ALGO.

While ALGO is evaluated with several searchers we proposed in the paper, there can be more possibilities for applying different searchers in the ALGO framework. For example, the self-refinement studies in large language models may be utilized for more complex interactions between the coder and the verifier. On the other hand, oracle-guided synthesis [14] is a well-studied topic in the programming language community with techniques that can be applied in different branches of code synthesis. We believe that knowledge from both the natural language processing community and the software engineering community can inspire more future work in the framework of ALGO.

## Acknowledgement

This work is partially supported by unrestricted gifts from IGSB and Meta via the Institute for Energy Efficiency.

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

# A  Appendix

## A.1  Examples of Algorithm Synthesis and Functionality Synthesis

We give examples of algorithm synthesis and functionality synthesis in Figure 8.

The upper part is an instance of algorithm synthesis, where input that describes the problem does not indicate the solution idea 'binary search' in any way. To solve the algorithm synthesis program, the model needs to come up with the idea related to 'binary search' first either implicitly or explicitly before synthesizing the code.

The lower part is an instance of functionality synthesis, where the input is basically a pseudo-code program described in natural language that can directly translate into Python code. The model does not need to come up with the idea because the solution is already clearly stated in the input.

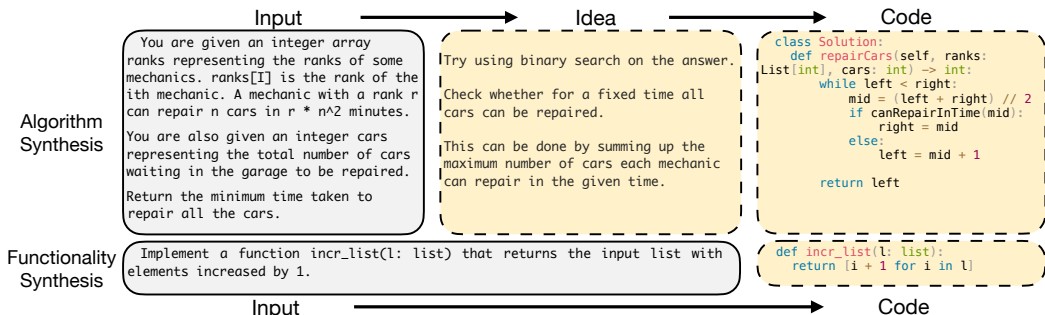

Figure 8: Examples of algorithm synthesis (top) and functionality synthesis (bottom). The parts in solid boxes are given by the problem, while the parts in dotted boxes should be inferred by the model. Functionality synthesis does not require the model to infer the idea, while algorithm synthesis does.

## A.2  Prompts in ALGO and Examples of Generated Programs

### A.2.1  LeetCode Examples

We list the prompts we used to generate the solution candidates and the verifier components for LeetCode problems here.

Figure 9 lists the prompts and examples for generating the naive solution. Figure 10 lists the prompts and examples for generating solutions guided by instructions that specify the algorithm category. Figure 11 lists the prompts and examples for generating reference oracle. Figure 12 lists an input validator example and its prompt. Figure 13 lists the prompt for the input generator that generates a single test input. Figure 14 lists the prompt for the batch input generator.

Figure 9: The naive prompt without instruction guidance and a candidate program generated with it. The instructions are in blue. It is a greedy method in linear time but is not suitable for this task. Even the naive candidate generated can pass the examples in the description, it fails both the system judge and ALGO verifier.

Figure 10: The prompt with instructions about the specific algorithm category (binary search). Binary search is the ideal algorithm used by human programmers for this task, with which the coder can generate both correct and efficient solutions. In general, right algorithm category can efficiently improve the quality of generated program.

Figure 11: The prompt we used for oracle generation and one oracle generated with it. The instructions are in blue. The language model is instructed to generate the most straightforward solution by enumerating over a very large search space of all combinations of relevant variables. The generated oracle enumerates all the possible ordered partitions of work allocations to find out the optimal one.

Figure 12: The prompt is utilized to generate a input validator to verify the validity of the generated test input, ensuring it aligns with the constraints specified in the description. In practice, this validation task is a functionality synthesis task, which can be easily solved by LLM.

### A.2.2 Codecontests Examples

We list the prompts we used to generate the solution candidates and the verifier components for Codecontests problems here. Figure 15 lists the prompts and examples for generating the reference oracle using an exhaustive search. Figure 16 lists the prompt for generating the batch input generator.

Figure 13: The prompt is used to generate a data generator, to generate extra test cases when combined with the reference oracle. In practice, this generator generation task is a functionality synthesis task, which can be easily solved by LLM.

Figure 14: The prompt is used to generate a batch generator that employs the already generated `gen_input` to generate several input cases by calling it multiple times.

Since the test set problems in Codecontests are all from `https://codeforces.com/`, they originally require standard input/output. However, we follow the setting in Codecontests by concatenating the entire standard input as a single string and asking the model to generate reference oracles that map a string to a string.

Figure 15: The prompt we used for oracle generation and one oracle generated with it. The instructions are in blue. The language model is instructed to generate the most straightforward solution by enumerating over a very large search space of all combinations of relevant variables.

## A.3 The List of Problems from LeetCode

We list the problems we collected from LeetCode as benchmarks to test ALGO in Table 5, among them are 10 easy problems, 18 medium problems, and 7 hard problems.

```python
def gen_input(n_max: int, m_max: int) -> str:
    n = random.randint(1, n_max)
    m = random.randint(1, min(m_max, 10 ** 6 // n))
    book_titles = set()
    while len(book_titles) < n:
        book_title =
''.join([random.choice(string.ascii_uppercase) for _ in
range(m)])
        book_titles.add(book_title)
    input_string = f'{n} {m}\n' + '\n'.join(book_titles)
    return input_string

def batch_gen_inputs(batch_size,) -> list:
    inputs = []
    for _ in range(batch_size):
        inputs.append(gen_input(5, 10))
    return inputs
```

Figure 16: The prompt is used to first generate a function `gen_input` and then generate a batch generator that employs the already generated `gen_input` to generate several input cases by calling it multiple times.

Table 5: The Leetcode problems we use. We only pick problems that were released concurrently or after GPT-4 to avoid contamination.

| Problem ID | Problem Name | Level |
|---|---|---|
| 2582 | pass-the-pillow | easy |
| 2583 | kth-largest-sum-in-a-binary-tree | medium |
| 2584 | split-the-array-to-make-coprime-products | hard |
| 2585 | number-of-ways-to-earn-points | hard |
| 2586 | count-the-number-of-vowel-strings-in-range | easy |
| 2587 | rearrange-array-to-maximize-prefix-score | medium |
| 2588 | count-the-number-of-beautiful-subarrays | medium |
| 2589 | minimum-time-to-complete-all-tasks | hard |
| 2591 | distribute-money-to-maximum-children | easy |
| 2592 | maximize-greatness-of-an-array | medium |
| 2593 | find-score-of-an-array-after-marking-all-elements | medium |
| 2594 | minimum-time-to-repair-cars | medium |
| 2595 | number-of-even-and-odd-bits | easy |
| 2596 | check-knight-tour-configuration | medium |
| 2597 | the-number-of-beautiful-subsets | medium |
| 2598 | smallest-missing-non-negative-integer-after-operations | medium |
| 2600 | k-items-with-the-maximum-sum | easy |
| 2601 | prime-subtraction-operation | medium |
| 2602 | minimum-operations-to-make-all-array-elements-equal | medium |
| 2603 | collect-coins-in-a-tree | hard |
| 2609 | find-the-longest-balanced-substring-of-a-binary-string | easy |
| 2610 | convert-an-array-into-a-2d-array-with-conditions | medium |
| 2611 | mice-and-cheese | medium |
| 2612 | minimum-reverse-operations | hard |
| 2614 | prime-in-diagonal | easy |
| 2615 | sum-of-distances | medium |
| 2616 | minimize-the-maximum-difference-of-pairs | medium |
| 2617 | minimum-number-of-visited-cells-in-a-grid | hard |
| 2639 | find-the-width-of-columns-of-a-grid | easy |
| 2640 | find-the-score-of-all-prefixes-of-an-array | medium |
| 2641 | cousins-in-binary-tree-ii | medium |
| 2643 | row-with-maximum-ones | easy |
| 2644 | find-the-maximum-divisibility-score | easy |
| 2645 | minimum-additions-to-make-valid-string | medium |
| 2646 | minimize-the-total-price-of-the-trips | hard |

