# OpenReview forum: "ALGO: Synthesizing Algorithmic Programs with Generated Oracle Verifiers"
_NeurIPS.cc/2023/Conference — NeurIPS 2023 poster_

### Official Review · Reviewer_oqZ5 · 2023-07-06

**Soundness:** 3 good
**Presentation:** 2 fair
**Contribution:** 3 good
**Rating:** 5
**Confidence:** 4

**Summary:**

This paper presents ALGO, a new method for code generation for LLMs. Given a problem description, ALGO first generates a brute-force implementation and dozens of test cases. Then, ALGO generates efficient implementations and ranks them by the agreement with the brute-force implementation on the generated test cases. The evaluation covers different LLMs, code generation techniques, and evaluation datasets. It shows that ALGO improves significantly over considered baselines.

**Strengths:**

The paper is easy-to-follow and has good experimental results. It also provides an interesting ablation analysis in Section 3.4.

**Weaknesses:**

### Terminology
Verification refers to proving formal properties of programs. The “verifier” fails to do so in two ways: (i) the “verifier” itself can often be wrong, and (ii) the “verification” is done on limited test cases and is thus unsound. I would suggest replacing “verifier” with a more appropriate term, such as cross-check, etc.

### Motivation
Lines 26-32 point out an issue of existing code generation approaches: lacking verifiability. However, I think ALGO suffers from the same issue, because the generated “verifier” can be wrong. The users still cannot have trust on the output of ALGO.

### Algorithm and functionality synthesis
I am not convinced by the difference between algorithm and functionality synthesis. The definition provided in the paper is very subjective and informal. For example, one can argue that the example of functionality synthesis in Figure 7 involves an idea of using a list comprehension instead of a for loop. In fact, both can be seen as standard text-to-code tasks. This makes me wonder how ALGO performs on HumanEval or MBPP.

### Discussion and comparison with existing work
The paper lacks a discussion and comparison with the following existing works. The first work also learns to “verify” generated code with test cases, while the second uses a reviewer to rank generated code.

LEVER: Learning to Verify Language-to-Code Generation with Execution. ICML 2023. https://arxiv.org/abs/2302.08468.
Coder Reviewer Reranking for Code Generation. ICML 2023. https://arxiv.org/abs/2211.16490.

### Cost analysis
It seems to me that ALGO can be significantly more expensive than the baselines. However, the paper does not include any cost analysis of ALGO.

### Instructions
What instructions do you use for the Instruction Enumerator at Line 137? It seems that the choices of instruction can significantly affect the performance.

### Small Issue
Line 55: InterpreterWe -> Interpreter. We


**Questions:**

Please consider addressing the points raised in the “Weakness” section.

**Limitations:**

While the paper includes a brief discussion of future work at Lines 359-365, it is very vague and does not pinpoint any limitation of ALGO. I believe that ALGO’s limitations, such as high cost, should be stated more clearly.

---

> ### Author Rebuttal · Authors · 2023-08-09
>
> Thank you for the thoughtful comments.
> ## 0. Terminology
> We understand your concerns. We will replace the term *verifier* in the revision. To avoid confusion, we use the terms "verifier" and "oracle" with quotation marks in the rebuttal.
> ## 1. Motivation and discussion about existing works
> We highlight that ALGO is a step forward towards verifiability which is fundamentally different from existing works like LEVER, Coder-Reviewer, and CodeT.
>
> **ALGO's verdict agrees well with the system judge.** The first three rows in Table 1 indicate that by using ALGO instead of CodeT as reranker on the exactly same set of program candidates, the top-ranked programs perform significantly better. Table 2 of our paper indicates that ALGO's verdict has a high agreement with that of the system judge.
>
> **ALGO is more reliable.** LEVER, Coder-Reviewer and CodeT checks programs only with language models. On the contrary, ALGO offloads the symbolic reasoning to a program. This technique makes models' reasoning more reliable and is used by other works such as PAL [1] and PoT [2].
>
> **ALGO is interpretable, existing works are not,** because they depend solely on neural models. ALGO provides test cases and "oracles" interpretable by human. If a user is not sure about ALGO's "verification", they can simply check if the brute-force "oracles" are right. When we manually checked "oracles" (Line 251 - 252), we found brute-force programs were much easier to check than actual solutions. In some sense, ALGO has converted the task of checking AI generated programs to checking AI generated brute-forces.
> ## 2. Comparison with existing work
> **LEVER has a different task setting, so it cannot be compared with ALGO.** To check a program, LEVER needs access to test-time system inputs. But ALGO doesn't. This makes things easier for LEVER since it only needs to focus on the correctness of this specific input. As they stated in limitations, LEVER is "ideal for applications such as text-to-SQL and math reasoning where the users are only looking for answers to their questions", but not as suitable for "general programming tasks as MBPP".
>
> **ALGO performs significantly better than Coder-Reviewer.** We used the officially released candidate programs of Coder-Reviewer. We replace their reranker with ALGO and keeping everything else the same. We report the results in the following table (same setting as Table 2 in Coder-Reviewer).
> ||Coder-Reviewer|CodeT|ALGO|
> |-|-|-|-|
> |pass@100|66.9%|65.8%|82.3%|
> ## 3. The scope of algorithm synthesis
> Here we define algorithm synthesis more formally and demonstrate its difference from functionality synthesis.
>
> **Algorithm synthesis.** Following the notations in Section 2.1, we use $J_S$ and $J_T$ to denote the system judges for semantic correctness and time efficiency. We denote all semantically correct programs with $P_S$, such that $J_S(P_i)=\mathsf{True},\forall P_i\in P_S$. $P_S$ can further be divided into two subsets $P_S^0$ and $P_S^1$, such that $J_T(P_i^0)=\mathsf{False},\forall P_i^0\in P_S^0$ and $J_T(P_i^1)=\mathsf{True}, \forall P_i^1\in P_S^1$. In other words, $P_S^0$ is correct but slow, while $P_S^1$ is correct and efficient. A task belongs to *algorithm synthesis* when finding a program within $P_S^1$ is more challenging than within $P_S^0$.
>
> **LeetCode and CodeContests are algorithm synthesis, while HumanEval is not.** In the following table, we report the success rates for ChatGPT to generate a program in $P_S^0$ and $P_S^1$ when prompted to. A brute-force is in $P_S^0$ if it is manually checked to be right. An efficient solution is in $P_S^1$ if it passes all system tests in time. For LeetCode and CodeContests, it is much easier to generate brute-forces than efficient solutions. But for HumanEval, it's similarly easy. Therefore, HumanEval is not algorithm synthesis according to our definition.
> ||LeetCode|CodeContests|HumanEval|
> |-|-|-|-|
> |$P_S^0$ (brute-force) succ. rate|88.5%|72.0%|70.12%|
> |$P_S^1$ (efficient) succ. rate|41.2%|7.9%|72.56%|
> |relative $\Delta$|114%|811%|-3%|
> ## 4. Cost analysis
> ALGO is NOT significantly more expensive than the baselines. We compare ALGO's cost with the baselines in two aspects - generation and validation.
>
> **ALGO has the same generation cost.** ALGO is a code generation framework where any coder can fit in. It is combined with existing code generation models like CodeX and ChatGPT. We compare ALGO to the baselines by keeping ALGO's sample budget to be the same. In our main experiments on CodeContests, ALGO is used to filter and rerank the exact same set of programs provided by CodeT. So it has the same generation cost.
>
> **ALGO has similar or cheaper validation cost.** Validation cost comes from model inference and test execution. For each problem, ALGO uses less then 5 model inferences to generate the "oracle" and the "verifier". However, the number of inferences for CodeT is proportional to the number of test cases it needs (which is 1000 for CodeContests), so ALGO costs much less in model inference. ALGO also costs less in test execution because ALGO can get much better performance with much fewer test cases (20 per problem) than the baseline (1000 per problem).
>
> We will include the above cost analysis in the revision.
> ## 5. Instructions
> The instructions we used are included in the supplementary files. We would like to emphasize here that while the instructions can significantly affect the performance, ALGO is able to improve the performance for different coders with different instruction sets (as demonstrated in our experiments).
>
> **Thank you again for your time. We kindly refer you to the global response for some common issues. Feel free to ask for further clarification.**
> ## References
> [1] PAL: Program-aided Language Models
>
> [2] Program of Thoughts Prompting: Disentangling Computation from Reasoning for Numerical Reasoning Tasks

---

> > ### Comment · Reviewer_oqZ5 · 2023-08-14
> >
> > I have read other reviews and the author rebuttals. I would like to thank the authors for providing clarifications, which addressed most of my concerns. Therefore, I raised my rating from 4 to 5.

---

### Official Review · Reviewer_MVRV · 2023-07-06

**Soundness:** 3 good
**Presentation:** 3 good
**Contribution:** 4 excellent
**Rating:** 7
**Confidence:** 4

**Summary:**

The paper introduces a new approach that utilizes Language Models (LLMs) to generate an oracle program for a given task, which is a brute-force search algorithm that is likely correct but potentially suboptimal in terms of runtime. The oracle program is then used to rank candidate programs based on their equivalence to the oracle on set of sampled inputs. The approach is applied to improve the performance of Codex and ChatGPT models and achieves additional improvements compared to the CodeT approach, which generates test cases on the fly and ranks programs based on consistency with those test cases. The proposed method is evaluated on challenging tasks from CodeContest benchmark and on a new set of LeetCode problems.


**Strengths:**

The paper presents an interesting idea by leveraging the fact that brute-force algorithm based oracles might be easy for LLMs to generate.
Moreover, compared to neural rankers or the CodeT approach, an oracle-based ranker presents a straightforward and reliable way to check for correctness of the sampled programs by just executing both programs on a set of sample inputs and checking the output consistency.
The evaluation includes challenging tasks from CodeContests, providing a robust assessment of the proposed method.

**Weaknesses:**

- The paper should mention estimates of inference times for all the approaches, including inference times for oracle generation, test generation, and code generation. It is important to discuss the tradeoffs associated with these inference times. For example, if CodeT is given the same inference time budget as ALGO, how does the performance difference change? Additionally, the impact of using an expensive and more capable model (ChatGPT) for oracle generation should be addressed, as it may introduce unfairness in the comparison with other approaches like CodeT.


- Please provide more details to enable reproducibility of the results. How many oracles need to be sampled before finding the correct one? What value of n is used for the various experiments? In Figure 4, what is the setup used for ChatGPT and GPT4? How many samples are drawn from each model? Are they are also sampled with different instructions? Including a Pass@k table for various k values would also be helpful for the experiment in Figure 4.


- It is unclear from the paper if the failure cases are due to the model's inability to generate any correct program within the n samples or if the oracle is not a good discriminator. Some analysis on whether the "top-ranked but incorrect programs" satisfy the oracle or not would help.



**Questions:**

Please see the weaknesses section for questions and suggestions.

**Limitations:**

Discuss the limitations regarding the increased inference cost of the proposed approach.

---

> ### Author Rebuttal · Authors · 2023-08-10
>
> Thank you for your valuable comments!
> ## 1 Inference cost estimation
> We argue that ALGO is NOT significantly more expensive than the baselines. We compare ALGO's cost with the baselines in two aspects - generation and validation.
>
> **ALGO has the same generation cost.** ALGO is a code generation framework where any coder can fit in. It is combined with existing code generation models like CodeX and ChatGPT. We compare ALGO to the baselines by keeping ALGO's sample budget to be the same. In our main experiments on CodeContests, ALGO is used to filter and rerank the exact same set of programs provided by CodeT. So it has the same generation cost.
>
> **ALGO has similar or cheaper validation cost.** Validation cost comes from model inference and test execution. For each problem, ALGO uses less then 5 model inferences to generate the "oracle" and the "verifier". However, the number of inferences for CodeT is proportional to the number of test cases it needs (which is 1000 for CodeContests), so ALGO costs much less in model inference. ALGO also costs less in test execution because ALGO can get much better performance with much fewer test cases (20 per problem) than the baseline (1000 per problem).
> ## 2 The impact of using a more capable model
> We provide extra baselines to show that ChatGPT is not the only reason why ALGO works. Instead of using ALGO's oracles to generate outputs, we directly used ChatGPT to generate the outputs of ALGO's test inputs.
>
> We use three different test sets to rerank ChatGPT candidates on CodeContests: ChatGPT + sample tests, ChatGPT + ChatGPT-generated tests, ChatGPT + ALGO tests. The sample budget for each problem was 20. For each problem we picked the top-ranked programs as its solution. We report the pass rates of the top-k-ranked programs (20@k) below.
> |  | 20@1 | 20@3 | 20@7 |
> |-|-|-|-|
> | ChatGPT + sample tests    | 4.4%    | 7.9%|12.2%|
> | ChatGPT + ChatGPT-generated tests | 6.8% | 8.2% | 11.7%|
> | ChatGPT + ALGO-generated tests |12.0%|12.0%|14.0%|
>
> As the table shows, even when the test generation module is replaced with ChatGPT, ALGO is still able to significantly improve code generation ability.
> ## 3 More details for reproducibility
> The sample budget for oracle generation was 3.
>
> The sample budget for CodeT was 1000. We directly took the candidates provided by CodeT without generating new candidates ourselves.
>
> The sample budget for PG-TD was 256, which follows the setting in the original paper.
>
> The sample budget for ChatGPT and GPT4 was 5.
>
> For pass rates with different top-k algorithms, we refer you to the table in the second part of this rebuttal *The impact of using a more capable model*.
>
> ## 4 When does ALGO fail?
>
> **Most of ALGO's failures come from the coder's inability to generate correct programs within n samples.**
>
> **This can be deduced from ALGO's performance gain on CodeContests.** For different coders, we use the same set of oracles and verfiers to guide their generation. For weak coders like Codex and GPT-2 (Table 1), ALGO's improvement on performance is significant even when k is as large as 100, indicating that as the sample budget gets larger and more correct solutions get generated, ALGO is able to discriminate the good from the bad. However, for a strong coder like ChatGPT (the table in *"The impact of using a more capable model"*), ALGO's improvement converges for a very small k, indicating that ALGO is already to accurately find the correct solution (even when only the top-1-ranked program is picked).
>
> **This can also be analyzed from the problems not solved by ALGO.** For the 7 hard-level problems in our leetcode dataset, ALGO's oracle was correct for 5 of them. However, only 1 of the 5 was correctly solved by at least one program. 4 out of 5 problems could have been solved and verified by ALGO if we had a stronger coder.
>
> Thank you again for your time! We kindly refer you to our global rebuttal for some common issues and some remarks about ALGO's application beyond algorithmic challenges.

---

> > ### Comment · Reviewer_MVRV · 2023-08-14
> > **Response to the rebuttal**
> >
> > Thanks for your response! It clarifies some of my questions. I appreciate the ChatGPT based baselines for the CodeContests dataset. I think they are very useful for a fair comparison. I would encourage the authors to do this experiment on a further extended set of this dataset (rather than just 50 problems).
> >
> > To create a fair comparison to CodeT, ideally, it would be nice to use the Codex model to generate the oracles in ALGO. But with Codex model being deprecated, it is not clear what is the right approach for this.

---

> > > ### Author Response · Authors · 2023-08-14
> > >
> > > Thank you!
> > >
> > > We really appreciate your suggestions and will conduct the experiments on the entire codecontests test set with 165 problems.
> > >
> > > We will try to create a fair comparison to CodeT without CodeX. One way we are considering is to replicate the CodeT process (test generation, program generation, and cross agreement) with ChatGPT as both the test generator and the program generator. We will include these results in the revision.

---

### Official Review · Reviewer_UrDq · 2023-07-07

**Soundness:** 3 good
**Presentation:** 3 good
**Contribution:** 2 fair
**Rating:** 6
**Confidence:** 4

**Summary:**

This paper presents ALGO which is a technique for generating programs for algorithmic challenges like LeetCode problems. To be successful, a code solution must be both correct and efficient enough. The core idea in ALGO is to first ask a code LLM to generate a slow brute force solution that is likely to be correct but too slow. This is used as an oracle verifier to check whether other program samples are correct for small test cases. Thus, an existing code generation system can propose solutions which are then filtered or reranked according to agreement with the verifier. The experiments show that ALGO’s technique of using a brute-force solution as a verifier leads to improvements on CodeContests and LeetCode.

**Strengths:**

Originality: A key observation is that current code LLMs can produce highly-accurate (but slow) brute-force solutions to tricky algorithmic questions. This was not previously obvious to me because one does not usually think about slow brute-force solutions. (I think the authors should emphasize this observation!) Then, assuming the existence of a slow but correct brute-force solution, the ALGO framework is a natural and straightforward application of using a verifier to aid program synthesis.

Quality: The experiments test a variety of ways of using ALGO with different models on different datasets.

Clarity: The paper is well organized and easy to read.

Significance: Improving code generation is certainly significant.

**Weaknesses:**

Originality: The process of using a verifier to filter/rerank solutions is pretty straightforward.

Quality: There are some concerns about comparing pass@k with reranking the top k among n samples. The experiments could also compare against other code reranking approaches on both CodeContests and LeetCode. See the Questions below.

Clarity: I have a few clarification questions below, but nothing major.

Significance: The paper only discusses this approach applied to algorithmic challenges, which has only moderate significance. In the Questions section below, I ask the authors to elaborate on how ALGO might be practically useful in other settings.

**Questions:**

## Pass@k given n samples

In line 216, what does “the same sampling budget $n$ under the same setting” mean? What is the “setting”? What is the actual value of $n$?

Is it really fair to compare pass@k given n samples, to a reranking approach returning the top k among n samples? If we increase n, pass@k does not change in expectation, but the top k approach would improve (assuming the reranking is accurate). For example, AlphaCode (https://arxiv.org/abs/2203.07814) distinguishes these metrics as “pass@k” vs “n@k” (but note that the AlphaCode paper swaps the meaning of n and k). These are different metrics that are not easily compared directly.


## Comparison to other reranking approaches

Can ALGO be compared to other code reranking approaches, such as CodeT and Coder-Reviewer (https://arxiv.org/pdf/2211.16490.pdf)? Then, all reranking approaches can use the same “k@n” metric -- whether a good solution exists in the best k among n samples. (I know CodeT is in Table 1, but I was hoping for a more extensive comparison.)


## CodeContests vs LeetCode

Some experiments are done with CodeContests and other experiments are done with LeetCode. Why are the main experiments (Table 1, Figure 4) not repeated for both CodeContests and LeetCode?


## Verifier correctness

It is unclear why the “time limit exceeded” (TLE) verdict is counted as “correct” for the verifier. I would agree that TLE means the verifier is “likely correct”, but it would be a leap to assert that all TLE verdicts are “correct” as a key result highlighted in the abstract and throughout the paper. If I’m not misunderstanding, then I think this claim should be rephrased or qualified.

The “88.5% correct” number (Fig. 2 and throughout the text) is only for LeetCode. Is it possible to get an analogous number for CodeContests?

Lines 251 - 253 mention manually verifying whether TLE solutions are functionally correct. What is the result of this? The next sentence says “88.5%” again, but this is exactly the AC + TLE amount from Fig. 2. Does this imply that you manually verified that *every* TLE solution was functionally correct? Please clarify.


## Generating inputs

Line 162 says “generate the input generator with an input validator” -- are there examples of this? In particular, the input generator and input validator are both *programs*, which is different from the CodeT approach of hallucinating inputs as text directly from the model, right? If so, this is a main difference that should be explained more thoroughly.

Generating valid inputs seems tricky, especially when there are problem-specific constraints that could be difficult or annoying to check (e.g., that a solution exists or is unique, that a graph is connected or contains no cycles, etc.). Can you comment on how often the input generator/validator programs work correctly?


## Applicability beyond algorithmic challenges

Outside of algorithmic challenges, can the authors comment on other situations were a code LLM can predict a verifier that is correct but undesirable for some reason, which then helps a synthesis search find better solutions? Maybe if the verifier is a lengthy inelegant solution and we want to search for shorter elegant solutions which rely on assumptions/heuristics/invariants that might or might not hold? Maybe the verifier is in a popular language but a solution is desired in an uncommon language?

Do you imagine the approach being applicable to improving efficiency of real-world code? If so, how would ALGO differ from existing approaches to that goal?

Having an example of something like this, even if not a full experiment, would make the paper seem much more applicable.


## Minor suggestions

* “Oracle” usually means something that is always correct. However, the oracles used in this paper are not actually always correct. The ALGO approach simply *assumes* they are correct and treats them *like* an oracle. This distinction was initially not clear to me and only became clear after reading further into the text. Please consider clarifying this in the abstract and introduction.
* “ChatGPT Code Interpreter” does not come with a citation, so it is extra important to clearly describe what it is, preferably when it is first mentioned (e.g., Line 55). The description in Lines 181 - 187 is good, just move it earlier.

* Line 55: missing space between “Interpreter” and “We”
* Line 82: What does it mean for a program to be “capable of solving” a problem? The program either passes the judge or not, so it either solves the problem or not.
* Line 84: “produce an program” -> “produce **a** program”
* Figure 3 caption: “polynomial time. While” -> “polynomial time, while”
* Line 127: “We use the variable $I$ to denote the **variable** that the coder **enumerates**” -- it is unclear what the bold “variable” refers to or how the enumeration works. Consider that the prompt from Figure 3 says “What do you think are some **variables** that may affect the answer and how do you think they can be **enumerated**?” but this is definitely a different kind of variable and enumeration. It gets confusing when the same terms are used to describe very different things.
* Line 145: “A iterative” -> “**An** iterative”
* Line 168: this sentence implies that the verifier can’t be arbitrary. Why not? Possibly the best verifier comes from Code Interpreter, but would the approach still work with a verifier generated by some other code LLM?
* Lines 231 - 233: “ChatGPT Code Interpreter is used as an instruction enumerator along with GPT-4’s performance” -- I don’t know what this means
* Line 322: “Oracle in Program Synthesis” -> “Oracle**s** in Program Synthesis”
* Line 322: “In traditional” -> “In **the** traditional”
* Line 325: “due to the ambiguity nature” -> “due to the **ambiguous** nature”
* Line 327: “in software testing domain” -> “in **the** software testing domain”
* Line 352: what does “similarity” mean here? Does it make sense to say “based on the ease of generating brute-force solutions to algorithmic problems”?
* Line 359: “paper. There” -> “paper, there”

**Limitations:**

The paper is missing a limitations section.

I think the main limitation is in the accuracy of the oracle solution. If the oracle solution is actually wrong, then the ALGO approach will hurt the overall synthesis performance. There doesn’t seem to be an easy way to identify when the oracle solution is wrong. Generating correct inputs also seems tricky and may be limited to problems with simple input formats and constraints.

Another limitation is in the general applicability of ALGO, since it only works on problems where an existing code LLM can produce (in only 1 sample) a functionally correct oracle solution.

---

> ### Author Rebuttal · Authors · 2023-08-10
>
> Thank you for your thoughtful comments and inspiring questions.
> ## 1 pass@k vs n@k
> **We followed CodeT's exact setting.** The "setting" in Line 216 means that we used ALGO to rerank the solution candidates (**n=1000** per problem) provided by CodeT, so our results is comparable to those in CodeT's Table 3.
>
> **CodeT directly compared pass@k and n@k,** because the pass@k for a coder without reranking is equivalent to the n@k for a coder whose reranker returns a uniformly random permutation. This is why CodeT "use the unbiased definition of pass@k as [their] baseline".
>
> **We understand your concern,** pass@k and n@k are not easily compared. But n@k's of two coders with reranking are, which is why we compared them in Table 1. We will clarify the difference between the two, and report pass@k only as metric for reference.
> ## 2 Originality and comparison to other reranking approaches
> We do not claim reranking as part of our novelty. We highlight that ALGO is fundamentally different from existing works in verifier quality. We compare ALGO with one more baseline, Coder-Reviewer here, and will include more extensive comparisons in the revision.
>
> **ALGO is more reliable.** LEVER, Coder-Reviewer and CodeT checks programs only with language models. On the contrary, ALGO offloads the symbolic reasoning to a program. This technique makes models' reasoning more reliable and is used by other works such as *PAL: Program-aided Language Models*.
>
> **ALGO is interpretable, existing works are not,** because they depend solely on neural models. ALGO provides test cases and "oracles" interpretable by human. If a user is not sure about ALGO's "verification", they can simply check if the brute-force "oracles" are right. When we manually checked "oracles" (Line 251 - 252), we found brute-force programs were much easier to check than actual solutions. In some sense, ALGO has converted the task of checking AI generated programs to checking AI generated brute-forces.
>
> **ALGO performs significantly better than Coder-Reviewer.** We used the officially released candidate programs of Coder-Reviewer. We replace their reranker with ALGO's and keep everything else the same. We report the results in the following table (same setting as Table 2 in Coder-Reviewer).
> ||Coder-Reviewer|CodeT|ALGO|
> |-|-|-|-|
> |pass@100|66.9%|65.8%|82.3%|
> ## 3 CodeContests vs LeetCode
> **We report the performance of ChatGPT and ChatGPT+ALGO on CodeContests (sample budget n=20).**
> ||k=1|k=3|k=10|
> |-|-|-|-|
> |ChatGPT,pass@k|4.4%|7.9%|12.2%|
> |ChatGPT+ALGO,20@k|12%|12%|14%|
>
> ALGO can achieve 12% even when k=1, indicating ALGO's superior ability to verify programs.
>
> CodeT cannot be run on LeetCode because its backbone CodeX is no longer accessible. Our experiments of PG-TD on LeetCode is still ongoing because it takes very long time. We will update the results as soon as possible.
> ## 4 Verifier correctness
> **We did not assert that all TLE verdicts were correct.** We did manually verify that *every* TLE solution was functionally correct. We believe the misunderstanding about this "assertion" is caused by our writing. We will rephrase and clarify this issue in the revision.
>
> **ALGO's verifiers are correct for 72% of CodeContests problems.** Due to the time limit we were not able to verify all 165 problems. To make the samples representative, we sampled 4 contests with a total of 50 problems from CodeContests to manually check them. 64% of all oracles get AC/TLE verdicts. **72%** of all oracles are functionally correct. The extra 8% comes from oracles that get runtime errors because their deep search recursions exceeded stack memory.
>
> ## 5 Generating inputs
>
> **All input generators/validators we used can be found** in supplementary files, along with the prompts we used to generate them. Examples of input generators and input validators can be found in the Appendix (Figures 11, 12, 13, 15).
>
> **ChatGPT can to produce input generators/validators that follow problem-specific constraints.** Some tricky but successful examples for codecontests (see supplemantary files) include: generating trees (1575I), primes (1603E), permutations (1622B). We provide two more examples by ChatGPT that generates bidirectional connected graphs and directed acyclic graphs.
> ```
> def generate_random_connected_graph(n, m):
>     if n < 2 or m < n - 1 or m > n * (n - 1) // 2:
>         return "Invalid n and m."
>     # Step 1: Ensure the graph is connected by creating a tree
>     edges = []
>     nodes = list(range(n))
>     random.shuffle(nodes)
>     for i in range(1, n):
>         edges.append((nodes[i - 1], nodes[i]))
>     # Step 2: Add random edges to the graph.
>     while len(edges) < m:
>         u, v = random.randint(0, n-1), random.randint(0, n-1)
>         if u > v:
>             u, v = v, u
>         if u != v and (u, v) not in edges:
>             edges.append((u, v))
>     # Convert to the desired format.
>     result = f"{n} {m}\n"
>     for edge in edges:
>         result += f"{edge[0]} {edge[1]}\n"
>     return result
> ```
> ```
> def generate_random_dag(num_nodes, num_edges):
>     if num_edges > num_nodes * (num_nodes - 1) / 2:
>         raise ValueError("Too many edges to form a DAG")
>
>     edges = set()
>     while len(edges) < num_edges:
>         u = random.randint(1, num_nodes - 1)  # ensure u < v
>         v = random.randint(u + 1, num_nodes)
>         edges.add((u, v))
>
>     result = f"{num_nodes} {num_edges}\n"
>     result += "\n".join(f"{u} {v}" for u, v in edges)
>     return result
> ```
> ## 6 Applicability beyond algorithmic challenges
> We are glad you asked this! The possibilities you mentioned are inspiring and captured the essense of our method: naive yet functionally correct programs that are easy to find.
>
> Due to the rebuttal's length limit, we kindly refer you to our global rebuttal, where we introduce an example of ALGO's application - SQL optimization and talk about other possibilities.
>
> ## 7 Minor suggestions
> Thank you for the detailed and comprehensive suggestions. We will address these in the revision.

---

> > ### Author Response · Authors · 2023-08-15
> >
> > Dear Reviewer UrDQ,
> >
> > We wanted to express our gratitude for the time and effort you have put into reviewing. Your comments and suggestions have been truly valuable in helping us refine our work.
> >
> > We have taken your feedback into consideration and submitted a detailed rebuttal addressing the concerns you raised. If possible, we kindly request that you take a moment to review our rebuttal and consider the clarifications and modifications we have presented.
> >
> > Thank you once again for your time and dedication. We are happy to address your further concerns.
> >
> > Authors

---

> > ### Comment · Reviewer_UrDq · 2023-08-17
> >
> > Thank you for the clarifications.
> >
> > Please do update the paper writing accordingly. In particular:
> > * Clarifying pass@k vs n@k and what the authors intend to compare
> > * Readers should not need to refer to another paper's tables to understand/contextualize this paper's results and comparisons.
> > * There's a big difference between: (1) the authors empirically finding that all TLE verdicts were always functionally correct, vs (2) TLE verdicts being correct as a principle
> > * Any new results discussed during the review process, and the GPS coordinates example from the global response (can be an appendix)
> >
> > **About input generators:** can you clarify the exact process used to create the input generators/validators? (I appreciate the inclusion of prompts in the supplementary material, but I'm hoping you can summarize for me.) **Was there any manual per-problem work**, for example: writing a problem-specific prompt describing problem-specific input requirements; manually inspecting the generators/validators to make sure they work as expected; re-sampling the input generator/validator when the first sample is known to be bad e.g. with a syntax/runtime error; or anything else? Are there any metrics (manual inspection, automated metrics based on running the generators/validators) to quantify **how often** the generators/validators are correct?
> >
> > If manual effort is needed to create the input generator on a per-problem basis, then this is a main weakness of the approach that must be discussed. If the input generator creation is entirely automated, then *sometimes* the process will fail, and we need an analysis of how often that happens and why.

---

> > > ### Author Response · Authors · 2023-08-17
> > >
> > > ## 1 Update the writing
> > > >Please do update the paper writing accordingly.
> > >
> > > Thank you for your suggestions. We will revise the paper according to the reviews and add new results emerged in the rebuttal period.
> > > ## 2 How we create input generators/validators
> > > >can you clarify the exact process used to create the input generators/validators?
> > >
> > > Yes. This process is entirely automated.
> > >  - First, the prompt template (as listed in `suppl/prompts.py`) is concatenated with the problem description (see `suppl/ALGO/leetcode_data/2582.md`) to obtain the complete prompt.
> > >  - Second, we feed the comlete prompt to ChatGPT and obtain its response, usually a mixture of code and text.
> > >  - Third, we take the last code snippet enclosed by backticks `` ``` `` from the response. If there is no backtick-enclosed code snippet in the resonse, we resample.
> > >
> > > Here we provide the validator prompt template and the problem description of Leetcode Problem 2582. They are concatenated to generate the validator.
> > >
> > > **Note that we do not manually write or put problem-specific input constraints in the prompt**. These constraints are part of the problem description. (see the example below.)
> > > ```
> > > You are given this leetcode problem. Please help me by generating a validator function `is_valid_input` that takes exactly the same inputs as the solution function and returns a boolean value indicating whether the input is valid and follows the constraints defined in the problem description. Please test your data validator by checking the validity of the example cases given in the problem description.
> > > ```
> > > ``````
> > > ### Pass the Pillow
> > >
> > > There are `n` people standing in a line labeled from `1` to `n`.
> > >
> > > (description omitted)
> > >
> > > Given the two positive integers `n` and `time`, return *the index of the person holding the pillow after* `time` *seconds*.
> > >
> > > **Example 1:**
> > >
> > > ```
> > > Input: n = 4, time = 5
> > > Output: 2
> > > Explanation: People pass the pillow in the following way: 1 -> 2 -> 3 -> 4 -> 3 -> 2.
> > > Afer five seconds, the pillow is given to the 2nd person.
> > > ```
> > >
> > > **Example 2:**
> > >
> > > ```
> > > Input: n = 3, time = 2
> > > Output: 3
> > > Explanation: People pass the pillow in the following way: 1 -> 2 -> 3.
> > > Afer two seconds, the pillow is given to the 3rd person.
> > > ```
> > >
> > > **Constraints:**
> > >
> > > - `2 <= n <= 1000`
> > > - `1 <= time <= 1000`
> > >
> > > **Function definition**
> > > ```
> > > class Solution:
> > >     def passThePillow(self, n: int, time: int) -> int:
> > > ```
> > > ``````
> > > >Was there any manual per-problem work?
> > >
> > > No, there wasn't, as stated in the paragraph above. The only resampling we did was when ChatGPT did not produce a program enclosed by back ticks `` ``` ``, which was completely automated.
> > > ## Generator/validator analysis
> > > >... we need an analysis of how often [creation failure] happens ...
> > >
> > > We manually checked the generators/validators of our Leetcode dataset and a subset (50 problems) of codecontests validation. We report the correct rates below. Note that for codeforces problems, we did not use data validators.
> > >
> > > |Dataset|correct generator%↑|correct validator%↑|
> > > |-|-|-|
> > > |Leetcode|100%|94.2%|
> > > |codecontests|96%|N/A|
> > >
> > > Correct rates for both datasets are well above 90%. For the rare incorrect generator/validators, we will explain **one by one** why their mistakes are minor and why they failed in the following analysis.
> > >
> > >
> > > >.. we need an analysis of ... why [creation fails]
> > >
> > > **For leetcode problems,** all generators are correct. Two problems got incorrect validators: 2643 and 2646.
> > >
> > > For 2643, the input is a matrix, but the validator doesn't check every subarray has the same length. However, this constraint is guaranteed by the test generator.
> > >
> > > For 2645, the input is a tree, but the validator doesn't check the edges in the input form a tree. However, this constraint is also guaranteed by the test generators.
> > >
> > > Therefore, the wrong validators in leetcode do not undermine the reliability of the generated inputs.
> > >
> > > **For code contests,** test generators are incorrect for only 2 problems - 1575B, 1575M.
> > >
> > > For 1575B, the generator doesn't guarantee the input has a solution. However, in this specific problem, inputs without solutions is hard to construct and did not appear in the test inputs generated at random. For other problems with requirements for solution existence, the test generators do guarantee it.
> > >
> > > For 1580E, the generator does not guarantee the input graph is connected. We suspect it's due to the descritpion of connectivity was not using graph theory terms, ("each railway station should be able to follow the rails and reach every other station"). This is a rare case, because other generator were able to guarantee connectivity when asked.
> > >
> > > Based on this analysis, we conclude that ALGO is very reliable in creating test generators and validators.
> > >
> > > ---
> > > Thank you again for your suggestions! We will put the clarification and analysis from this response in the revised paper, along with others. We are happy to clarify if you have further questions.
> > >
> > > If you find that we've addressed your concerns, we kindly ask you to reconsider your score.

---

> > > > ### Author Response · Authors · 2023-08-19
> > > >
> > > > Hi Reviewer UrDQ,
> > > >
> > > > Since the rebuttal period is coming to an end, we'd like to know if we've addressed all your concerns. If there are any additional points you'd like to discuss, we are more than happy to respond.
> > > >
> > > > Best,
> > > > Authors

---

> > > > > ### Comment · Reviewer_UrDq · 2023-08-21
> > > > >
> > > > > My concerns were adequately addressed. I will increase my score from 5 to 6, due to the satisfactory answers to all of my questions and extra details/experiments.
> > > > >
> > > > > One final comment: I suspect the paper's good results come largely from the power of ChatGPT Code Interpreter. Specifically, the ALGO method relies on the high accuracy of generated brute-force solutions and input generators/validators. If you were to use the same general approach but without ChatGPT Code Interpreter (e.g., using plain ChatGPT or GPT-4 instead), would the approach still outperform prior methods? If not, then this would be an important limitation to mention: that the ALGO approach relies on the powerful code generation ability of Code Interpreter. This is not that big of a limitation since code generation abilities should only improve in the future, but it is important context to know whether the ALGO approach would have worked before Code Interpreter was released.

---

> > > > > > ### Author Response · Authors · 2023-08-21
> > > > > >
> > > > > > Thank you for recognizing our efforts and the final comment!
> > > > > >
> > > > > > It makes sense to compare CodeInterpreter with plain ChatGPT/GPT-4. Due to the time limit of the author-reviewer discussion period, we could not conduct an oracle generation experiment with plain ChatGPT/GPT-4 in time. We will include these experiments in the revision.
> > > > > >
> > > > > > **We argue that plain ChatGPT/GPT-4 can also do a good job in oracle generation for the following reasons.**
> > > > > >
> > > > > > First, the ChatGPT Code interpreter we used in the paper (`text-davinci-002-code`, which is now deprecated) is essentially GPT-3.5 with code execution feedback. Its code generation ability is similar to that of the underlying model, GPT-3.5. We can implement a wrapper so that plain ChatGPT can also consider the code execution feedback. In our LeetCode experiment, GPT-4 actually performed better than CodeInterpreter because GPT-4 is stronger than GPT-3.5.
> > > > > >
> > > > > > Second, most "oracles" are generated in the first code snippet, while the execution feedbacks from CodeInterpreter benefit the subsequent snippets. Therefore, ChatGPT might perform well in oracle generation even without execution feedback.
> > > > > >
> > > > > > We will include this discussion along with more experimental results in the revision.
> > > > > >
> > > > > > Thank you again for your thoughtful comments! They have really helped us in refining our paper.

---

### Official Review · Reviewer_MLQT · 2023-07-07

**Soundness:** 2 fair
**Presentation:** 3 good
**Contribution:** 3 good
**Rating:** 5
**Confidence:** 5

**Summary:**

The paper tries to improve code generation ability. The main idea of the paper is as follows

1. Use an LLM to generate a brute force solution to contest coding problems -- the idea being that the hard part for such problems is not solving them but solving them given the time and memory constraints.

2. Have an test case input generator for the problem and get the corresponding output for the test case using the brute force solution in (1)

3. Verify whether a solution is correct using the input/output pairs generated in (1) and return the failing test cases to the code synthesizer

They test their algorithm using CodeContests and LeetCode and using multiple code synthesizers showing significant performance improvements.

**Strengths:**

A simple but nice idea leveraging the insight that the hard part for contest coding is solving the problem in the time and space constraints. They do a good set of experiments demonstrating the performance improvements. They also have additional analysis on the quality of their brute force solutions (valid 88% of the time for LeetCode)

**Weaknesses:**

The main worry I have with this paper is the following :-

*Baselines*

They use ChatGPT to generate the oracle verifier and show that using it can improve the performance of *weaker* models like Codex and GPT-Neo. However for CodeContests (their primary benchmark), they do not give the baseline of how good ChatGPT itself is at solving the problem.

This is an issue as it is expected that having a correct verifier is going to enable the selection of the best program out of a list of generated programs. In practice however, we will not have the luxury of having the correct verifier. We will have an LLM and the question before us would be whether generating a verifier using a strong LLM and using it to guide weaker LLMs (or the same strong LLM) is actually better than directly trying to use the strong LLM to generate a solution. Thus that baseline is critical but absent.

For LeetCode dataset, there is a ChatGPT baseline. However key experimental details are missing. In particular

a. What code synthesizer was used for ALGO?
b. If it was ChatGPT with CodeInterpreter, was the ChatGPT baseline and GPT-4 matched in terms of sampling budget? (If selection of the the best answer out of multiple answers is an issue, then a strategy like CodeT can be used) What was the sampling budget?


In addition, there's a simple baseline that could be tried that is not present -- what if you asked ChatGPT to generate the test input/outputs directly instead of generating a brute force solution?


*Temperature of generation*

There seems to be a temperature mismatch between CodeT and Algo w/ Codex. The results available online for the former from which the authors say they took their results are with temperature 0.8 whereas the authors results for Algo are with temperature 1. This could be putting the former method at a disadvantage as it has lower diversity in the generations.  In addition the temperature used for the other models is not mentioned. If the model does not have a filtering mechanism, then a low or 0 temperature should be used as otherwise the method is going to be at a disadvantage


**Questions:**

In addition to questions in the Weaknesses section above, I also have the following questions -

1. I can't find the value used for n (sampling budget) in the paper or the number of inputs generated using the oracle. That should be mentioned in the experimental section.

2. How good are the oracle solutions for CodeContests?

---

> ### Author Rebuttal · Authors · 2023-08-10
>
> Thank you for the thoughtful comments.
>
> ## 1 ChatGPT baselines on CodeContests
> We ran three experiments with ChatGPT as the coder on CodeContests: ChatGPT + sample tests, ChatGPT + ChatGPT-generated tests, ChatGPT + ALGO tests. The experiments were conducted on a 50-problem subset of CodeContests. The sample budget for each problem was 20 and we ranked the generated programs with different sets of tests. For each problem we picked the top-ranked program as its solution. We report the pass rates of the top-k-ranked programs (20@k) below.
> |  | 20@1 | 20@3 | 20@7 |
> |-|-|-|-|
> | ChatGPT + sample tests    | 4.4%    | 7.9%|12.2%|
> | ChatGPT + ChatGPT-generated tests | 6.8% | 8.2% | 11.7%|
> | ChatGPT + ALGO-generated tests |12.0%|12.0%|14.0%|
>
> **The results indicate that ALGO is able to significantly improve upon ChatGPT, and ALGO's test cases are better at discriminating correct solutions from incorrect ones.**
> ## 2 Clarification for sample budget, temperature, and number of tests
>
> **Sample budget.** In the original experiments on LeetCode, ChatGPT and GPT-4 was given a sample budget of 5. Since ALGO was using instruction enumerator, each instruction was given a sample budget of 1 and the "verifier" was used to pick the best program among those generated from different instructions. We understand your concern that this may cause unfair comparison, so we re-ran the experiments by giving the baselines a larger sampe budget equal to the size of the instruction set (10). We report the new results in the following table. **ChatGPT+ALGO is still the best after we increase other baselines' sample budget.**
>
> |  | pass rate |
> | -------- | -------- |
> | ChatGPT (budget = 5) pass@1    | 37.1%     |
> | GPT-4 (budget = 5) pass@1 | 45.7% |
> | ChatGPT + ALGO (budget = 1 per instruction) 10@1 | **48.6%** |
> | ChatGPT (budget = 10) pass@1 | 39.6% |
> | GPT-4 (budget = 10) pass@1 | 46.4% |
>
> **Temperature.** There is no temperature mismatch between ALGO+Codex and CodeT. Codex, CodeT and ALGO + CodeX in Table 1 are all using the same set of CodeX-generated candidate programs provided by CodeT. The only difference is how these programs were reranked. The temperature 1.0 we mentioned on Line 160 was during the generation of the "oracles", not the solution programs. We will make it clear in the revision.
>
> **Number of tests.** For the main experiments (in Table 1 and Figure 4), the number of tests generated by ALGO was 20.
>
> ## 3 Sample budget for oracle generation
> ALGO used a sample budget of 3 to generate oracle for each problem.
>
> ## 4 "Oracle" quality for CodeContests
>
> Due to the time limit, we sampled 50 problems from the CodeContests test set to manually check the correctness of the generated "oracles". **It turned out that 72% of them were correct.**
>
> We thank you again for your time and kindly remind you to take a look at our global rebuttal and supplementary files if needed.

---

> > ### Comment · Reviewer_MLQT · 2023-08-11
> >
> > Thank you for the results. I'm increasing my score to a 5.
> >
> > Another experiment I would suggest that the authors include is where the generated oracle verifier is used as part of the context for generating the more efficient method

---

### Author Rebuttal · Authors · 2023-08-10

We thank the reviewers for their thoughtful reviews and comments. We would like to address some common issues mentioned in multiple reviews and clarify the main points of our paper. For reviewer-specific issues, we have responded in reviewer-specific rebuttals.
## 0. Reminders
**Terminology.** We understand several reviewers' concerns about the proper use of the terms *oracle* and *verifier*, and promise to use more appropriate alternates in the revision. To avoid confusion in the rebuttal, we will refer to the exhaustive search programs generated by language models as "oracles", and the program correctness checker as "verifiers", with quotation marks, throughout the rebuttal period.

**Supplementary files.** We kindly remind the reviewers that the characteristics of the "oracles" and "verifiers" we talk about can be examined in the supplementary files, which contain our prompts, input generators/validators, "oracles", test cases and candidate programs.
## 1. ALGO's "verifier" is fundamentally different from existing works.
We thank the reviewers for pointing out some existing works such as LEVER [1] and Coder-Reviewer [2]. We will cite them and discuss more in the revision. Here we argue that ALGO's "verifier" has fundamental differences compared to the existing works. These differences come from its neurosymbolic nature.

**ALGO provides more reliable feedback.** Neural "verifiers" [1,2] provide little feedback by only giving a "correctness" score. Previous test-based "verifiers" like CodeT [3] provide some feedback with their LM-generated test cases, whose correctness solely relies on the arithmetic and symbolic reasoning ability of language models. However, language models often make mistakes in arithmetic and symbolic reasoning, which is why they solve math problems better by generating programs rather than directly outputting the answer, as demonstrated in PAL [4] and PoT [5]. Therefore, the test cases generated by LMs for a SINGLE problem probably contain both correct and incorrect ones, providing harmful feedbacks to the coder. However, once we have a correct "oracle" for a problem (which we do for 88% Leetcode problems), ALGO guarantees that ALL test cases generated will be correct.

**ALGO is interpretable and easy to validate.** Unlike LEVER and Coder-Reviewer which use inexplainable neural models to score the correctness of a program, ALGO uses "oracles" in the form of exhaustive search programs. For a candidate program, there is no way of telling how a neural model comes to its "correctness" score and whether the score actually reflects correctness. But any programmer is able to tell if a brute-force "oracle" is correct. According to our hired human annotators, manually checking a brute-force program is much easier than checking an efficient one using advanced algorithms. Because efficient solutions are diverse and problem-specific, while brute-force programs are similar and mostly enumerating permutations, subsets and stepwise choices.

## 2. ALGO may be applicable to general algorithm synthesis scenarios beyond programming contests.
In this part, we address reviewers' doubts about the scope of *algorithm synthesis* and if ALGO can be applied beyond programming contests. In general, ALGO can be applied to scenarios where a naive, correct but less satisfactory oracle is much easier to find then a satisfactory solution. Language models can be used to find this naive solution, which can later be used to guide the search for a satisfactory solution.

**Now we give a concrete example on how ALGO can assist LLMs to using programming languages and frameworks that aren't well-represented in the training data.** Here, the naive yet less satisfactory oracle is a program that's written in a popular language/framework well-represented in the training set.

Consider a basic function that fetches a city's name from Google Maps API using GPS coordinates. This program needs to run on an embedded system with limited memory and storage, therefore requiring it to be written in C++ using the `neon` library.

**Directly prompting GPT-4 won't work.** It got us the following code snippet.

**[First link to code snippet that we sent to the AC]**

When we tested the above code using LLM-generated test cases, the generated cases were not correct.

```
User: Generate test cases for a program that fetches city name based on GPS coordinates.
Input: 33.798242689239636, 2.87416689773028
Output:

ChatGPT: Input: 33.798242689239636, 2.87416689773028
Expected Output: City Name: Al Qadarif
```

In the provided example, ChatGPT incorrectly recognizes the location as Al Qadarif, whereas it's actually Laghouat, a city 2000 miles away.

Instead of hinging upon LLM’s ability to map inputs to outputs, ALGO can develop a straightforward "verifier" utilizing popular solutions like Python's requests library to ensure its accuracy. We presented the GPT-4 generated "oracle" in Python below.

**[Second link to code snippet that we sent to the AC]**

The above "oracle" can then check and guide the generation of a more efficient solution in C++.

There are other scenarios where ALGO may be applied, including loop optimization, computation graph scheduling and parallel computing.

We would like to thank the reviewers and the chairs for their time and efforts!

## Reference
[1] LEVER: Learning to Verify Language-to-Code Generation with Execution

[2] Coder Reviewer Reranking for Code Generation

[3] CodeT: Code Generation with Generated Tests

[4] PAL: Program-aided Language Models

[5] Program of Thoughts Prompting: Disentangling Computation from Reasoning for Numerical Reasoning Tasks

---

> ### Author Response · Authors · 2023-08-11
> **The two links to the code examples**
>
> The following two links are the example code we mentioned in the rebuttal - gpt-generated C++ and python code.
>
> [1] https://pastebin.com/T8FVH3x3
>
> [2] https://pastebin.com/uHuevXAg

---

### Decision · Program_Chairs · 2023-09-21

**Decision:**

Accept (poster)

**Comment:**

This submission presents ALGO, a method for solving programming challenges. The challenges are usually made up from a natural language task description and some test cases. Correctness is judged based on (held-out) test cases and often imposes running time limits.
The presented idea is to first use a prompt asking the LLM to generate a simple, “brute-force” solution, independent of running time concerns. This is often easy in algorithmic programming challenges. Then, the LLM is also asked to generate an input generator, whose outputs are used to compare the “brute-force” solution and potential efficient solution. Failures can be used to inform the generation of an improved efficient solution.

Reviewers highlighted that the proposed idea is simple but insightful (MLQT, MVRV) and that the experiments are well-suited to illustrate the utility of the proposed method (MLQT, UrDq, MVRV, oqZ5). Concerns about details of the experiments were cleared up during the rebuttal phase.

However, reviewers voiced doubts about the applicability of the proposed method to the general setting (oqZ5, UrDq), as the basic assumption about the ease of generating brute-force solutions seems to only hold in the programming challenge setting. While the authors provided a further example in the global rebuttal (though in their reply to UrDq also referred to a SQL optimization example, which does not appear?), that example does not seem to support the hypothesis of generalizing to the more general setting. The “simplistic” and “full” solutions do not differ in algorithmic complexity - they are effectively the same, and the C++ variant fails because GPT-4 hallucinates libneon interfaces, not because of an algorithmic mistake. Fixing this just requires a model better at writing C++, and the "simplistic" solution as oracle doesn't really help.

Overall, this is a borderline case. The authors have made a convincing case for their idea in the competitive programming setting, but so far there's little hard evidence that the method would be useful in the more general program synthesis case. Nonetheless, I think this may inspire interesting follow-up work and hence should be presented to a wider audience at NeurIPS.